# Pooled library screening with multiplexed Cpf1 library

Jintan Liu [1,2], Sanjana Srinivasan[1,2], Chieh-Yuan Li[1,2], I-Lin Ho[1,2], Johnathon Rose[1,2], MennatAllah Shaheen[1,2], Gang Wang[3], Wantong Yao[1,6], Angela Deem[4], Chris Bristow[4], Traver Hart[3,5] & Giulio Draetta [1,2]

Capitalizing on the inherent multiplexing capability of AsCpf1, we developed a multiplexed, high-throughput screening strategy that minimizes library size without sacrificing gene targeting efficiency. We demonstrated that AsCpf1 can be used for functional genomics screenings and that an AsCpf1-based multiplexed library performs similarly as compared to currently available monocistronic CRISPR/Cas9 libraries, with only one vector required for each gene. We construct the smallest whole-genome CRISPR knock-out library, Mini-human, for the human genome ($n = 17,032$ constructs targeting 16,977 protein-coding genes), which performs favorably compared to conventional Cas9 libraries.

[1] Department of Genomic Medicine, The University of Texas MD Anderson Cancer Center, Houston, TX 77030, USA. [2] MD Anderson Cancer Center UTHealth Graduate School of Biomedical Sciences Houston, Houston, TX 77030, USA. [3] Department of Bioinformatics and Computational Biology, The University of Texas MD Anderson Cancer Center, Houston, TX 77030, USA. [4] Institute for Applied Cancer Science, The University of Texas MD Anderson Cancer Center, Houston, TX 77030, USA. [5] Department of Cancer Biology, The University of Texas MD Anderson Cancer Center, Houston, TX 77030, USA. [6] Present address: Department of Translational Molecular Pathology, The University of Texas MD Anderson Cancer Center, Houston, TX 77030, USA. Correspondence and requests for materials should be addressed to J.L. (email: Jliu14@mdanderson.org) or to G.D. (email: GDraetta@mdanderson.org)

High-throughput forward genetic screens are an invaluable tool to systematically explore genetic interactions and to link gene disruption with disease contexts[1]. The adaptation of CRISPR/Cas9 has resulted in pooled-library knockout screens with improved sensitivity and specificity versus pooled-library shRNA screens, primarily owing to more limited off-target effects, more potent gene perturbation, and a higher proportion of active constructs in such systems compared to an equivalent shRNA pool[2–4].

Cas9-based pooled-library screens have demonstrated their feasibility and efficacy in numerous studies[1,3,5]. Like pooled shRNA screens, pooled sgRNA screens benefit from multiple constructs targeting each gene of interest, as the functionality of each construct is largely unknown. However, increasing the number of genetic reagents per target to improve the efficacy of the library comes at the expense of increasing library complexity, which in turn increases labor requirements as well as costs for reagents and sequencing. Because inactive constructs cannot be physically removed from the library, they serve as confounding factors during hit identification. While this can be partially mitigated with hits identification algorithms such as MAGeCK[6] and BAGEL[7,8], there remains a need to further enhance CRISPR technology to improve library penetrance with smaller, more versatile pooled libraries. Several algorithms have been developed to aid in the design of sgRNAs with high efficiency guides[4,9–11] to minimize the number of constructs needed to yield robust results, but there remains a need for additional advancements to vastly decrease library size.

Similar to CRISPR/Cas9, CRISPR/Cpf1 (CRISPR from Prevotella and Francisella 1, or Cas12a) is a class 2 CRISPR system identified in the prokaryotic adaptive immune system that cleaves target DNA using much shorter RNA guides compared with Cas9[12]. It has been demonstrated that the Cpf1 orthologues LbCpf1 and AsCpf1 are highly specific, comparable or greater than SpCas9[13,14]. Moreover, Cpf1 is self-sufficient for multiplexed gene editing, unlike Cas9, which requires other Cas proteins and RNase III to process polycistronic guide precursors in its native host[15]. Although modified Cas9 multiplexing systems, such as tRNA[16], Cas6/Csy4[17], ribozyme aided[18] and tandem expressing cassettes of sgRNAs[19–22] have been created, the long scaffold sequence and long repetitive elements between sgRNA guides result in vector recombination and template shuffling, which significantly cripple their utility in functional genomics screens, where most data are acquired using short-read next-generation sequencing.

Based on its inherent properties, including short guide length and autonomous guide processing activity, we inferred that the CRISPR/Cpf1 system may enable multiplexed libraries compatible with pooled-library screenings. The system would eliminate the need for the trade-off between library efficacy and library complexity by multiplexing of different guides targeting the same gene into a single lentiviral vector.

However, even though Cpf1 is active in mammalian cells, multiple groups have reported diminished efficiency compared with SpCas9[14,23,24], and whether Cpf1 offers an adequate solution for the purpose of genome screening remains unknown. Here we show that with optimization, AsCpf1 can be used for functional genomics screenings and that an AsCpf1-based multiplexed library performs similarly as compared to currently available mono-cistronic CRISPR/Cas9 libraries, with a significantly reduced library complexity. We construct the smallest whole-genome CRISPR knockout library, Mini-human, for the human genome ($n = 17,032$ constructs targeting 16,977 protein-coding genes), which performs favorably compared to conventional Cas9 libraries.

## Results

**Design and construction of benchmark CRISPR libraries.** To assess the performance of Cpf1 multiplexing, we generated a multiplexed AsCpf1 library targeting 342 core-essential genes and 345 non-essential genes, with three guides per gene[3,4] (2061 guides, 687 constructs). Fitness changes upon knockout of these genes are highly consistent across multiple cell lines, therefore making them gold-standard controls. To compare the screening performance of the multiplexed AsCpf1 library and conventional mono-cistronic CRISPR knockout libraries, we generated two additional benchmark CRISPR libraries targeting the same group of genes: an SpCas9-based mono-cistronic library (2061 guides, 2061 constructs) and an AsCpf1-based mono-cistronic library (2061 guides, 2061 constructs). The design rules for SpCas9 and AsCpf1 guides are highly similar, despite nuclease-specific requirements, such as different protospacer adjacent motifs (PAMs). The AsCpf1-based mono-cistronic and multiplexed libraries shared identical guide sequences; however, the multiplexed AsCpf1 library had only a single construct harboring all three guides (Fig. 1a).

We noted extensive rearrangements between the multiplexed AsCpf1 guide arrays when using Gibson assembly to clone synthesized oligo pools into the backbone. The rearrangements included loss of guides and random shuffling of guides (Supplementary Fig. 1a, b, c). This phenomenon was independent of the polymerases used in the PCR reaction as well as the bacteria host strains (Supplementary Table 1). However, the rearrangements were not observed when we employed the classical restriction enzyme-based cloning, suggesting that the identical sequence of AsCpf1 scaffolds might have led to incorrect recombinations during Gibson assembly (Supplementary Fig. 2).

**Optimization of AsCpf1 for functional genomics screening.** For these benchmark screens, we employed K-562 cells separately infected with each pooled CRISPR library at a low multiplicity of infection (MOI). After puromycin selection, we delivered the corresponding CRISPR nuclease (AsCpf1 or SpCas9) by lentivirus transduction and blasticidin selection. We conducted triplicate screens with cells grown for 4 weeks and sampled each replica at intermediate time points to capture the dynamics of guide populations. We measured screen performance by the relative differential fold change of core-essential vs non-essential genes (Fig. 1b).

Somewhat unexpectedly, the common AsCpf1 variant used (human codon-optimized AsCpf1 with a C-terminal nucleoplasmin bi-partite nucleus localization signal (NLS), herein AsCpf1-Nuc)[13,25] failed to show satisfactory separation of essential vs non-essential genes in the screen (Supplementary Fig. 3a), despite showing activity when tested by T7E1 assay with individual constructs from the libraries (Supplementary Fig. 3b). We hypothesized that this might be caused by inefficient AsCpf1 nuclear translocation. To test this hypothesis, we evaluated library performance with one of several AsCpf1 variants carrying different NLSs. We found that an AsCpf1 bearing N-terminal 3x MYC-NLS and an optimized Kozak sequence (herein AsCpf1–3xMYC) effectively discriminated positive and negative control genes in our benchmark library screens (Supplementary Fig. 3a, c). Moreover, when cells were infected by AsCpf1–3xMYC or Aspf1-Nuc at identical MOI, AsCpf1–3xMYC showed stronger expression and nuclear localization compared to AsCpf1-Nuc (Supplementary Fig. 3d). Using the T7E1 assay, we also confirmed a much higher editing efficiency for this variant compared to AsCpf1-Nuc (Supplementary Fig. 3e), suggesting that the amount of AsCpf1 in the nuclear fraction is critical for optimal gene editing efficiency.

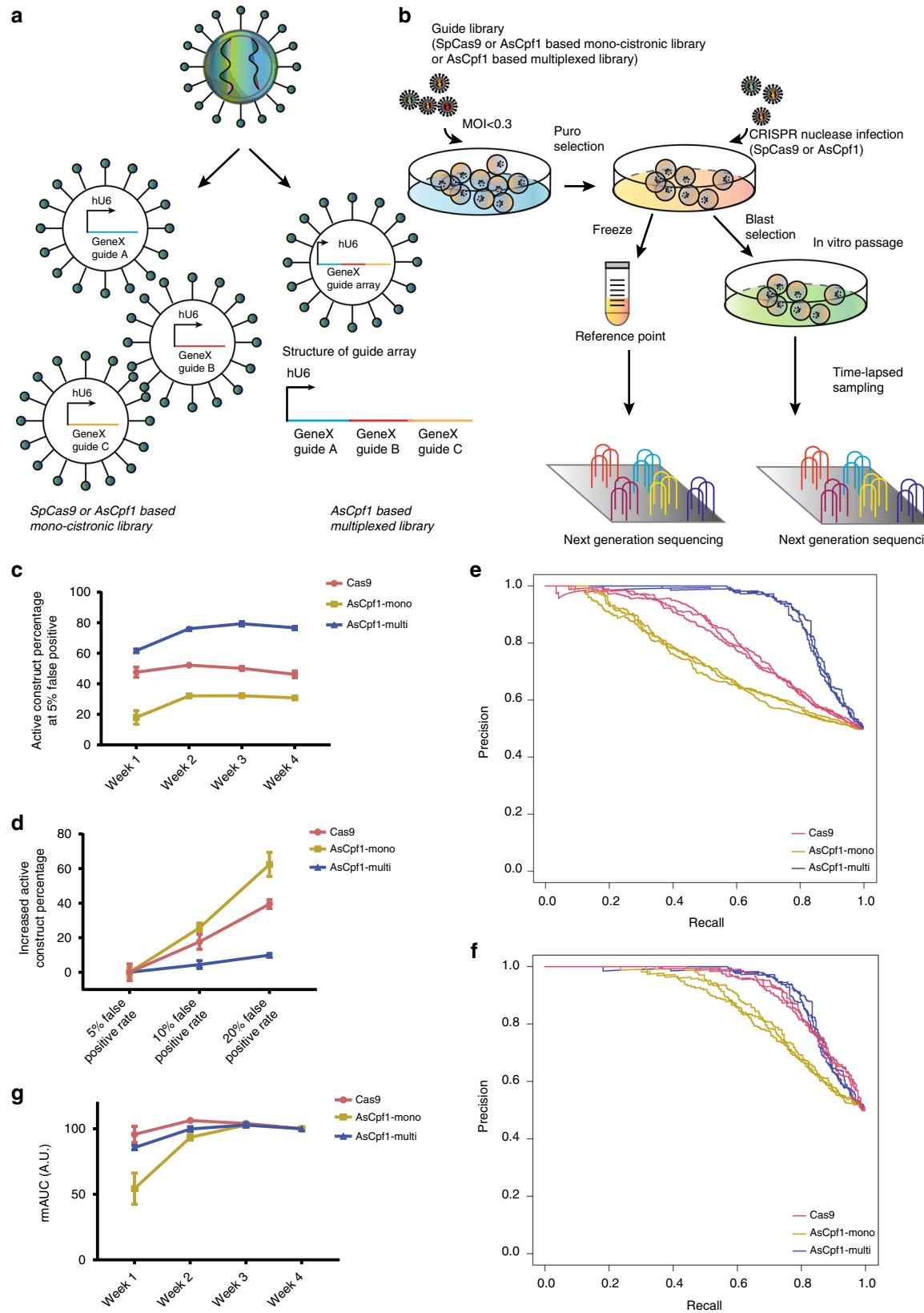

**Functional genomics screening with benchmark libraries**. We then repeated the benchmark screens with AsCpf1–3xMYC. All biological replicates for each of the three library screens were well correlated, indicating good reproducibility (Supplementary Fig. 4a, 4b,4c). An essential gene-targeting construct was considered active if it was more depleted compared to a non-essential gene-targeting construct since it should have an anti-proliferative effect if active. To determine the percentages of active constructs among the three different libraries, we chose a false positive rate (FPR) of 5% based on the log2 transformed fold change for each time point. The active construct percentage curve of the SpCas9-based mono-cistronic library was relatively flat across all four

**Fig. 1** Functional genomics screenings with benchmark CRISPR libraries **a** In SpCas9 or AsCpf1-based mono-cistronic libraries, each gene has multiple guides dispersed in different lentivirus. In AsCpf1-based multiplexed library, each gene only has one guide array construct. However, each array contains multiple different guides targeting one gene. **b** Pooled library screen pipeline schematics. MOI (Multiplicity of infection) **c, d** The percentage of the active construct for different libraries at the 5% false positive rate across four time points (**c**), and the increased percentage of the active construct for different libraries at the screen endpoint with a different controlled false positive rate (**d**). Pink: Cas9-based mono-cistronic CRISPR library (Cas9). Yellow: AsCpf1-based mono-cistronic CRISPR library (AsCpf1-Mono). Purple: AsCpf1-based multiplexed CRISPR library (AsCpf1-Multi). Error bars are present as standard deviations (s.d.) of triplicate. **e, f** The construct-level precision-recall curves for CRISPR libraries (**e**), and the gene-level precision-recall curves for CRISPR libraries (**f**). Each curve represents one biological replicate, three replicates in total. Pink: Cas9-based mono-cistronic CRISPR library. Purple: AsCpf1-based multiplexed CRISPR library. Yellow: AsCpf1-based mono-cistronic CRISPR library. **g** The rmAUC curves of CRISPR libraries describing population dynamics. rmAUC (ratio of the modified area under the curve) is calculated by $(AUCx-0.498)/(AUCend-0.498) \times 100\%$. $AUCx$: the area under the curve of construct-wise precision-recall curves for time point $X$. $AUCend$: the area under the curve of construct-wise precision-recall curves of an endpoint. Pink: Cas9-based mono-cistronic CRISPR library. Yellow: AsCpf1-based mono-cistronic CRISPR library. Purple: AsCpf1-based multiplexed CRISPR library. Error bars are present as s.d. of triplicate. Source data are provided as a Source Data file

time points, with a mean value of 49.0% ± 2.9% active constructs. For AsCpf1-based libraries, the active construct percentage curve plateaued 2 weeks after the screen initiated with mean values of 31.7% ± 0.9% and 77.4% ± 1.4% active constructs for the mono-cistronic and multiplexed libraries, respectively (Fig. 1c). The different shapes of the active construct percentage curves for SpCas9- and AsCpf1-based screens indicate different population temporal dynamics and knockout efficiency for the different CRISPR nucleases.

Our data suggest that SpCas9 is more active in mammalian cell gene knockout experiments compared to AsCpf1, consistent with prior reports[14,23,24]. However, multiplexing different guides targeting the same gene significantly increased the likelihood of gene knockout with AsCpf1. At endpoint, the percentage of active constructs in the AsCpf1 multiplexed library was only slightly increased compared to other libraries when we relaxed the FPR stringency from 5% to 20% (increased by 4.4% ± 1.9% at 10% FPR and 9.9% ± 0.8% at 20% FPR) (Fig. 1d), highlighting the high signal-to-noise ratio of the multiplexed AsCpf1 screen.

To call out significantly depleted genes, we used an adapted Bayesian Analysis of Gene Essentiality (BAGEL) algorithm to analyze construct-level data. On the basis of the fold change in sgRNA abundance after knockout of each gene in the essential and non-essential training sets, BAGEL uses a Bayesian model selection approach to classify a Bayes Factor (BF), which is the log2 likelihood of each gene belonging to either the essential gene distribution or non-essential gene distribution. Because BAGEL is designed for whole-genome CRISPR screens, we designed and utilized a version of BAGEL optimized for small library screens, Low Fat BAGEL. Low Fat BAGEL generates BFs on a construct-level basis that is summed across guides to obtain a gene-level BF. For the AsCpf1-based multiplexed library, each gene has only one corresponding construct; therefore, its construct-level BF corresponds directly to its gene-level BF.

To benchmark screen performance across the three libraries, precision-recall curves were plotted based on BFs. The precision-recall curves clearly showed that the SpCas9-based mono-cistronic screen (construct-wise area under the curve (AUC) 0.78 ± 0.01, gene-wise AUC 0.89 ± 0.01) outperformed the AsCpf1-based mono-cistronic screen (construct-wise AUC 0.70 ± 0.01, gene-wise AUC 0.82 ± 0.01) at both the construct (Fig. 1e) and the gene level (Fig. 1f). However, the AsCpf1-based multiplexed screen (construct-wise and gene-wise AUC 0.89 ± 0.00) performed similarly to the SpCas9 mono-cistronic library at the gene level, and it yielded a much stronger performance at the construct level, primarily due to lower active construct percentage in the SpCas9 screen.

To compare the separation rate between essential and non-essential genes among the three library screens, we calculated the ratio of the modified area under the precision-recall curve

(mAUC) of any given time point divided by the mAUC at endpoint (ratio of mAUC, rmAUC) (Fig. 1g). As the area under the precision-recall curve for this library would be 0.498 when there is no separation between essential and non-essential genes, the mAUC for any given time point was set to be its AUC minus 0.498. As shown in Fig. 1c, the results suggest that separation between essential and non-essential genes in the AsCpf1-based mono-cistronic screens was much slower compared to SpCas9-based mono-cistronic and AsCpf1-based multiplexed screens. This might be the result of the relatively slower cleavage dynamics of AsCpf1 compared with SpCas9[26], as we also saw a slightly slower separation between essential genes and non-essential genes in the AsCpf1-based multiplexed screen compared with SpCas9 indicating the slower separation is not unique to the mono-cistronic AsCpf1 library.

Overall, the multiplexed AsCpf1 and SpCas9 libraries performed comparably, while the mono-cistronic AsCpf1 library was inferior to both. To compare the performance at the individual gene level, we performed a gene-matched Wilcoxon test based on the rank of essential genes across the three libraries. We failed to find statistically significant differences ($p = 0.2409$ for Cas9 vs Cpf1 mono-cistronic, $p = 0.2142$ for Cas9 vs Cpf1 multiplexed, Wilcoxon signed-rank test, two-sided), indicating that Cas9- and Cpf1-based libraries performed similarly at the individual gene level. We also performed Spearman's correlation analysis on the z-score of essential genes, and found significant correlations between Cas9- and Cpf1-based libraries (Spearman's $Rho = 0.48$, $p = 2 \times 10^{-40}$ for Cas9 vs Cpf1 mono-cistronic; Spearman's $Rho = 0.57$, $p = 2 \times 10^{-61}$ for Cas9 vs Cpf1 multiplexed, Spearman's rank-order correlation, two-sided), again demonstrating similarly high performance of Cas9- and Cpf1-based libraries (Supplementary Fig. 5a, 5b, 5c).

**AsCpf1's guide design optimization.** CRISPR/SpCas9 guide design has been optimized using empirical data from hundreds of screens[4,6,9,27], but previous AsCpf1 guide optimization algorithms were largely based on a small number of surrogate reporter experiments[25]. It is known that lentiviruses have integration site biases[28] and that the chromosomal environment can influence CRISPR nuclease activity[29]. Thus, the gene editing process on surrogate reporters might not fully represent the true biological effects on endogenous loci editing. Our screen provides a large-scale action-in-situ dataset to enable prediction of AsCpf1 guide preference based on functional screening data of endogenous loci. We used fold change information of essential gene-targeting guides in the mono-cistronic AsCpf1 library to calculate sequence preference, as effective guides should drop out more efficiently than ineffective ones. Even though all of the 342 genes are considered essential genes, the severity of their knockout phenotypes may differ. To avoid sequence biases introduced by

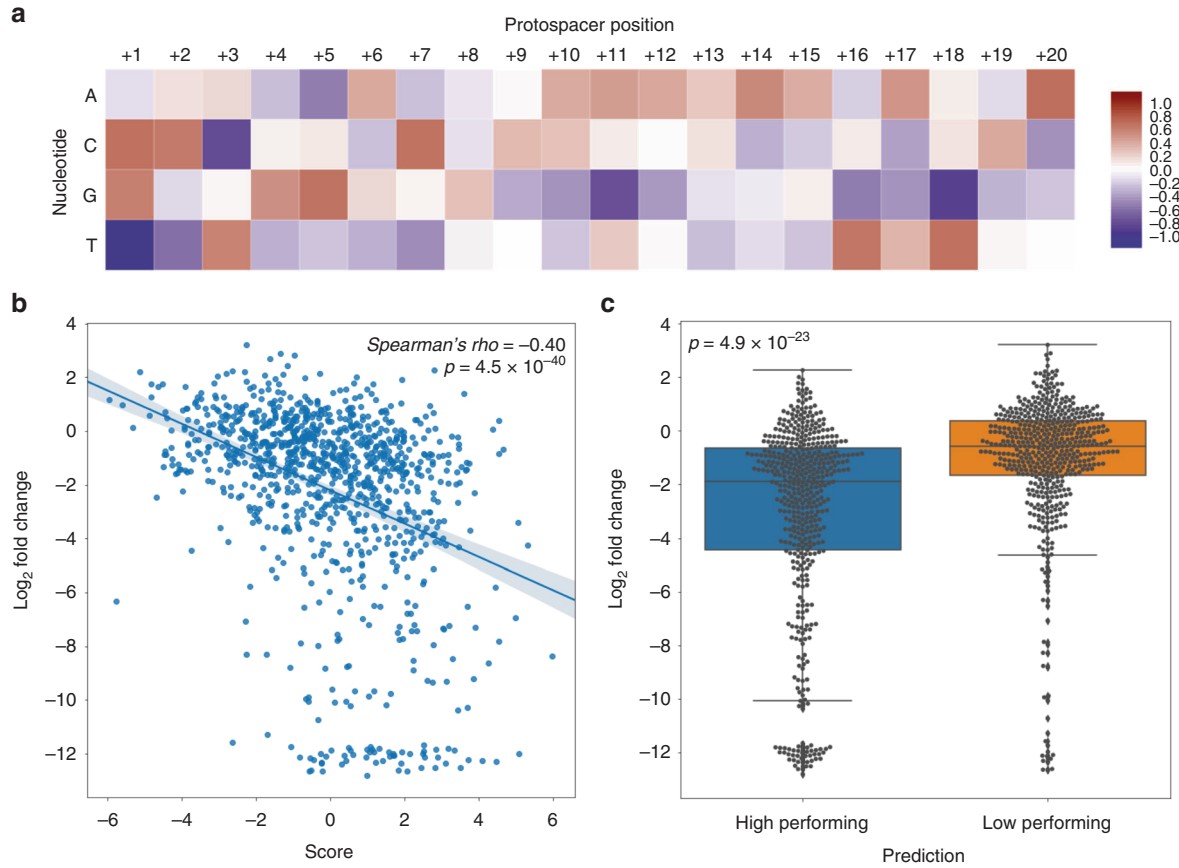

**Fig. 2** Sequence preference features of AsCpf1 **a** The scoring matrix for the prediction of AsCpf1 activity on endogenous loci. A positive value (in red tone) indicates a preference towards a nucleotide for a high-performing guide while negative value (in blue tone) indicates a preference against a high-performing guide. The position of the first nucleotide adjacent to PAM is defined as position +1 for the protospacer. **b** Correlation between AsCpf1 guide prediction scores and depletion levels of core-essential genes in the AsCpf1-based mono-cistronic library screenings. The predicted AsCpf1 guide scores for each guide in the validation set across 100 bootstrappings were plotted against their log2 transformed fold change in the screen (n = 342). Line of best fit is plotted and the shaded area is presented as s.d. **c** Box plot of predicted high-performing AsCpf1 guides (HighPerforming) and low-performing guides (LowPerforming) across 100 bootstrappings. Guides were defined as high-performing if predicting score >0 and low-performing if predicting score ≤0 (n = 342). Centerlines are present as medians. Bounds of boxes are present as first and third quantiles and whiskers are present as minimums and maximums without outliers. Source data are provided as a Source Data file

gene-specific effects, within the three guides in each gene, we classified the most depleted guide as the high-performing guide and the least depleted guide as the low-performing guide. Using a scoring scheme similar to that described by Hart et al.[4], we calculated the frequency of each nucleotide at each position of the 20-mer protospacer individually for high-performing and low-performing guides. At each position, the nucleotide frequency of low-performing guide was subtracted from the high-performing guide to produce a table with subtracted frequencies for each nucleotide. This process was repeated across 100 bootstraps, and an aggregate average score table for each protospacer position was obtained. In agreement with a previous report[25], thymine (T) was strongly disfavored in position 1 in the protospacer, while guanine (G) and cytosine (C) were favored. We also identified features not previously reported. For example, we found a general trend of G disfavor from position +16 to position +20 in the protospacers. In addition, we found that T was favored in positions +3, +16, and +18, while C was favored in position +7 but disfavored in position +3 (Fig. 2a). The score table obtained served as a metric to predict guide activity in terms of fold change: the indicated sequence score for each guide represented the sum of the nucleotide score at each position. Therefore, a guide with a zero-sequence score indicates no similarity between either effective or non-effective guides. We validated our

prediction algorithm on the median-performing guides not used to develop the scoring algorithm—that is, the guide for each gene that was neither the highest nor lowest performing guide. Each median-performing guide was assigned a sequence score and was classified with a prediction of high-performing or low-performing based on a guide score >0 or ≤0, respectively. We then evaluated if the sequence score and predicted performance classification of each guide were indicative of fold change and found a significant correlation between the guide score and guide performance (Spearman's rho = −0.40, 95% confidence interval: (−0.34, −0.45), p = 4.5 × 10⁻⁴⁰, Spearman's rank-order correlation, two-sided) (Fig. 2b). The predicted high-performing guides were significantly more effective than low-performing guides, with a mean log2 fold change of −3.19 compared to −1.16 (t = −10.13, p = 4.9 × 10⁻²³, student t-test, two-sided) (Fig. 2c).

**Genome-wide screening with AsCpf1-based multiplexed library.** Based on our multiplexed AsCpf1 library strategy, we designed and constructed the smallest available CRISPR library targeting the entire human protein-coding genome, Mini-human. The guides for Mini-human were optimized based on activity scores derived from our screen dataset and further filtered for potential off-target effects. Because a previous analysis of published screens determined that for SpCas9, four to six gRNAs per

gene yield robust results when computational approaches to design sequence-optimized guides are employed[4,6,27], each construct in Mini-human contained up to four gene-targeting guides: 16393 gene-targeting constructs with four optimized guides, 584 gene-targeting constructs containing three optimized guides, and 55 non-targeting guide arrays as negative controls. This library was approximately one-fourth the size of the smallest currently available genome-wide CRISPR library and will be made publicly available.

We conducted a genome-wide screen using the Mini-human library (17,032 constructs) in K-562, a chronic myelogenous leukemia cell line, using conditions similar to other published screenings in this model using the GeCKOv2 (123,411 constructs) and the gold standard Wang library (the Sabatini dataset; 187,536 constructs). At the pathway level, GSEA analysis identified common pathways related to cell survival and replication depleted across all three datasets, including those related to the spliceosome, ribosome, and DNA replication (Supplementary Figs. 6, 7, 8). To compare the datasets at the level of hits, we conducted a BAGEL analysis with a cutoff threshold for hit identification set to a false discovery rate (FDR) of 0.01. A similar number of hits was identified by Mini-human and Wang library, followed by GeCKOv2, and the overlap percentages of hits between any two libraries was similar for all hits sets and core-essential gene hits sets (Fig. 3a, b). In accordance with recent reports, each library identified unique hits[30]. We performed gene ontology enrichment analysis for dataset specific hits, which failed to identify any significantly enriched pathways. Taken together, our results support that SpCas9 and AsCpf1 libraries induced similar cell proliferative behaviors and both are suitable for functional genomics applications.

Next, we used the gene-level BF generated precision-recall curves to enable a comparison of overall library performance. We determined that the Sabatini dataset outperformed both datasets, and Mini-human outperformed GeCKOv2 (AUC = 0.98, AUC = 0.96, and AUC = 0.94, respectively) (Fig. 3c). The number of guides per gene differs among all three libraries: Mini-human has 3–4 guides per gene on a single construct, whereas the GeCKOv2

and Wang libraries use 6 or 10 mono-cistronic guides per gene, respectively. The Wang library employed an optimized guide design strategy, and even after down-sampling (average of 10 random down-samplings with four guides per gene), it still outperformed the other libraries (Supplementary Fig. 9). Overall, the libraries performed similarly in the K-562 functional screening, indicating that a multiplexed AsCpf1 library can yield robust results comparable with commonly used SpCas9-based mono-cistronic libraries in this context.

**Genotoxicity with AsCpf1-based multiplexed genome editing.** Multiple groups have reported copy number-related genotoxicity in SpCas9-based pooled-library screenings wherein highly amplified genes showed reduced fitness regardless of their biological function. In the multiplexed AsCpf1 library, each lentivirus particle is capable of introducing multiple double-strand breaks (DSBs) into a single locus. To investigate whether these on-target DSBs may similarly diminish fitness, we randomly selected four non-essential genes (*ADAM18, IL3, PAX4,* and *VSX2*) and compared their multiplexed AsCpf1 constructs from Mini-human and their corresponding mono-cistronic guides. Cells containing either mono-cistronic guides or multiplexed guide arrays were transfected with an AsCpf1-expressing plasmid and were subjected to biochemical analysis. Analysis of cell viability via CCK-8 assay identified no difference between cells infected with multiplexed versus mono-cistronic vectors (Fig. 4a). Furthermore, H2A.X phosphorylation levels measured by In-cell Western analysis were similar between the two groups, whereas a control culture of cells treated with 100 nM gemcitabine for one hour accumulated phosphorylated H2A.X, as expected (Fig. 4b). Finally, quantification of Annexin-V was similar between cells harboring multiplexed or monocistronic vectors (Fig. 4c). Thus, the multiplexed guide vectors in Mini-human do not negatively impact cell viability or induce accumulation of DNA damage or apoptosis compared to monocistronic guide vectors, and our data indicate that the deleterious effects of copy number-related, locus-independent cutting by SpCas9 are not recapitulated in AsCpf1-based multiplexed systems.

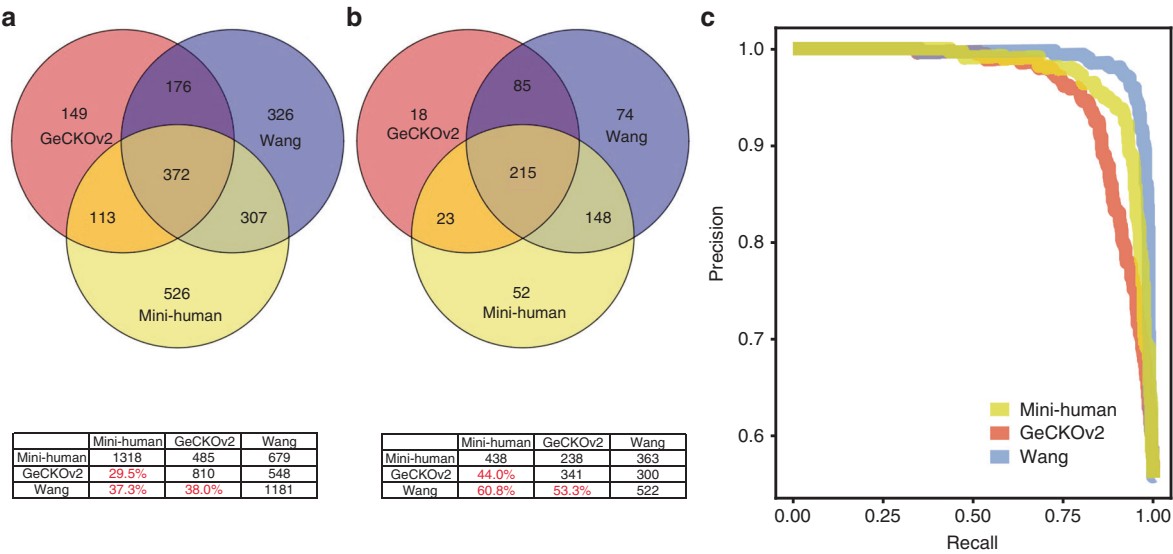

**Fig. 3** Performance of Mini-human genome-wide knockout library. **a** Hits detected by three different libraries (GeCKOv2, Wang and Mini-human) were compared. The overlapping numbers were represented in the Venn diagram while the hits concordance rates (in red) of any two libraries were represented in the table below. *FDR (False discovery rate)* threshold: 0.01 **b** Overlaps and concordance rates of known core-essential genes identified as hits by three different libraries. *FDR* threshold:0.01 **c** The gene-level precision-recall curves for genome-wide CRISPR libraries. Red: GeCKOv2 library. Yellow: Mini-human library. Blue: Wang library

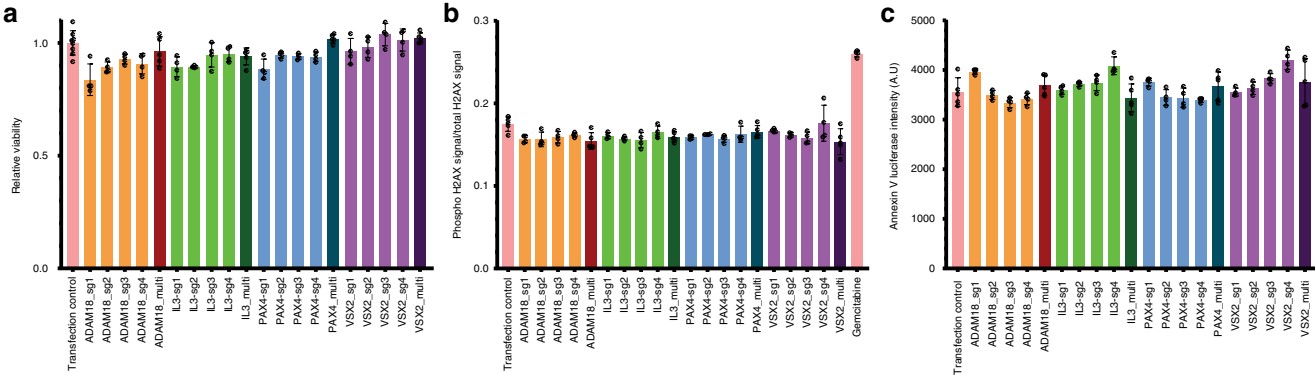

**Fig. 4** Genotoxicity with AsCpf1-based multiplexed genome editing Cells were transfected with AsCpf1–3xMYC plasmid together with either mono-cistronic guides or multiplexed guide arrays targeting non-essential genes. Each multiplexed guide array contains four different mono-cistronic guides used in this study. For example, ADAM18_multi contains 4 different guides ADAM18_sg1, ADAM18_sg2, ADAM18_sg3, and ADAM18_sg4. **a** Relative viability of cells treated by AsCpf1 and different single guide vectors or multiplexed guide vectors. $n = 8$ for transfection control, $n = 4$ for all mono-cistronic vectors, $n = 5$ for all multiplexed vectors. Error bars are present as s.d. **b** Phosphorylated H2A.X signal of cells treated by AsCpf1 and different single guide vectors or multiplexed guide vectors. Gemcitabine: 1-hour 100 nM gemcitabine treatment prior to In-cell Western assay as a positive control. $n = 4$ for all mono-cistronic vectors, $n = 5$ for all multiplexed vectors and transfection control. Error bars are present as s.d. **c** Annexin-V staining signal of cells treated by AsCpf1 and different single guide vectors or multiplexed guide vectors. $n = 4$ for all mono-cistronic vectors, $n = 5$ for all multiplexed vectors and transfection control. A.U. arbitrary units. Error bars are present as s.d. Source data are provided as a Source Data file

## Discussion

We provide evidence to support the deployment of a multiplexed, AsCpf1-based system for functional genomics applications. In accordance with previous reports, we found that, in general, AsCpf1 underperforms SpCas9 in terms of activity, and we found that AsCpf1's performance correlated with its nuclear localization. However, we demonstrate a significant improvement in AsCpf1-based library performance when guides were multiplexed into a single vector. We thus created a library that combines AsCpf1–3xMYC, a variant with optimized nuclear localization and cutting, with multiplexed guide vectors, and demonstrated that it performed similarly to conventional SpCas9 libraries. Our experiments provide proof of principle supporting the utility of multiplexed AsCpf1-based libraries to address a critical need in functional genomics for highly efficient pooled libraries with low complexity.

It is not clear why multiplexing guides enhances the performance of AsCpf1-based libraries. One likely contributing factor is that the probability of generating an on-target loss-of-function mutation is increased, because of better sampling of an effective guide from the multiplexed vector, thereby reducing the screening analysis noise introduced by nonfunctional constructs. It is also possible that, by introducing more than one cut on a single locus, the multiplexed library may increase the chances of a functional deletion. Specifically, the multiple cuts may delete a larger nucleotide sequence compared with the indels that are the dominant repair product at a single endonuclease cleavage site. Consistent with this idea are reports that bi-cistronic SpCas9 guides can generate fragment deletions[31].

We used our benchmark mono-cistronic AsCpf1 screening dataset to generate preference rules for designing guides for AsCpf1. This is the first dataset available to mine for AsCpf1 guide preference on endogenous loci and may reveal chromatin features that are not detected by surrogate reporters. Our data are consistent with previous reports that uridine in the first spacer nucleotide position (thymine in the first protospacer position) has a strong, negative effect on guide performance. We also uncovered previously unappreciated preferences for guanine and cytosine at specific positions. These features are previously unidentified by surrogate reporters and are not biases from selecting the core-essential gene guides as the training dataset, as there is

no significant correlation between guide scores and depletion level for non-essential genes (Spearman's rho = -0.01, $p = 0.65$, Spearman's rank-order correlation, two-sided) (Supplementary Fig. 10). Whether these features are cell-specific or if they are affected by chromatin configuration remain interesting points for future investigation.

No obvious locus-independent cytotoxic effect was identified for the multiplexed AsCpf1 guide vectors. The high signal-to-noise ratio of the multiplexed library indicated there is no interference in the pooled-library screen using the multiplexed guide strategy. Biochemical assays using several non-essential gene vectors also confirmed no additional cytotoxicity caused by multiplexing. This was in contrast with the reported on-target cytotoxic effect of SpCas9 targeting high copy number regions. This could be explained by differential sensitivity to DSBs of different cell lines, or it is feasible that four DSBs are not sufficient to trigger locus-independent cytotoxicity. Another possible explanation is that the genomic distances between DSBs introduced by multiplexed AsCpf1 vectors are different compared to SpCas9-induced DSBs that have been shown to exert a copy number-related cytotoxic effect. Over 90% of the multiplexed AsCpf1 guides span a genomic distance <10 kb (Supplementary Fig. 11), which is smaller than the scale of common copy number amplifications. In scenarios where guides span larger distances, one potential approach to mitigate concerns about DSB-induced toxicity is to sample early time points for a reference population, as copy number effects are reported to appear early in the screening process, or apply scoring algorithms such as CERES[8] to take the copy number effect into consideration. The amplification of off-target effects by multiplexed vectors can be addressed by selecting guides that are predicted to be low off-targeting or off-target free, which is feasible for AsCpf1 and LbCpf1, for which off-target properties have been extensively studied[13,14]. Moreover, the higher fidelity of Cpf1 compared with SpCas9 contributes to mitigating the increased risk of off-target effects introduced by multiplexing. Whether multiplexing alone will introduce de novo off-targets that are not seen in the off-target union set of individual guides remains unknown. The higher order structural changes of guide arrays caused by multiplexing might change RNA post-transcriptional machinery and RNA stability, therefore changing the reagent's fidelity. Alternatively, for cell lines that are

sensitive enough to respond to the additional DNA damage caused by multiplexing, a changed DNA damaging response might lead to a different off-target profile of the same guide.

Using catalytically dead Cpf1 (dCpf1) fusion proteins for CRISPRi or CRISPRa screens is another option to avoid on-target toxicity in knockout screens. Cpf1 has been shown to induce or suppress gene expression when fused with other functional domains[32–34], and there are reports of a strong, on-target synergistic effect of multiplexed Cpf1-based CRISPRi and CRIS-PRa strategies[35].

We constructed the first multiplexed genome-wide CRISPR knockout library, Mini-human, which represents a new, powerful tool for demanding functional genomics applications, especially in vivo pooled-library screenings where library size is a concern. Using K-562 cells as a model system, we demonstrated that Mini-human performs similarly well compared with conventional SpCas9-based libraries such as GeCKOv2 and the Wang library. The superior performance of the Wang library can be attributed to its much larger number of guides per gene (10 guides per gene vs 6 for GeCKOv2 and 3–4 for Mini-human) and the high coverage used in the screening (1000× vs 500× for GeCKOv2 and Mini-human). These data suggest that increasing guide number within the multiplexed vector may also be beneficial in the context of AsCpf1-based multiplexed libraries. Current oligo pool synthesis technologies enable multiplexing of up to seven guides, with the upper limit resulting from technical factors surrounding the synthesis of oligo pools. With a carefully designed cloning strategy, a larger number of multiplexed guides may be attainable.

For AsCpf1, the number of guides available for each gene can sometimes be limiting for library construction. AsCpf1 has a strict requirement for the TTTV PAM motif, resulting in a smaller selection of guides available compared with SpCas9, which requires NGG or NAG PAM. Multiple AsCpf1 variants have been identified that can target non-canonical PAMs such as TYCV, TATV, TTYN, VTTV, and TRTV. Using these variants instead of wild-type could increase the guide selection pool for AsCpf1-based libraries; however, it is unknown whether the variants' protospacer nucleotide preferences and off-target properties may vary versus wild-type. It was reported[15] that there is no position effect when AsCpf1 guide arrays were individually tested. In the context of a pooled-library screening, expression of the guide array will be more vulnerable to various cellular machinery since only one copy of array expression cassette will be found in most infected cells. Determining whether a guide's efficiency is influenced by its cistron position in a pooled-library screening will be important to further optimize AsCpf1 multiplexed library strategies.

It is reasonable to assume that a combinatorial gene knockout/down screen is feasible using multiplexed Cpf1 guide vectors. High-throughput screening (HTS) for multi-gene perturbation effects could greatly facilitate functional interaction studies, such as identifying synthetic lethality to position cancer therapeutics. Previous efforts to develop HTS systems have been based primarily on RNAi and SpCas9. While, theoretically, RNAi can be multiplexed, it has inferior precision compared with SpCas9. Moreover, multiple pairs of shRNA or SpCas9 guide expression cassettes must be tested independently to determine the effect of perturbing a gene pair of interest. For SpCas9, a maximum of two guides can be multiplexed into an expression cassette if direct sequencing of guide spacers is desired. Furthermore, paired SpCas9-based screenings suffer from unwanted recombination from vector construction and sequencing library preparations from PCR. By contrast, AsCpf1 guides are shorter, and AsCpf1 possesses autonomous guide processing capabilities that enable a single, multiplexed vector to deliver constructs targeting multiple genes. This approach greatly reduces library complexity. As an example, to probe all random two-gene combinations of 1000 genes using six perturbagens (guides or shRNAs) per gene, SpCas9 and RNAi would require a 36 million-vector library. In stark contrast, a multiplexed AsCpf1 library would require only 1 million vectors. Furthermore, because AsCpf1 libraries can use 300 bp, pair-end sequencing to multiplex up to 13 guides, AsCpf1 libraries could enable multi-gene (>2) interactions to be probed. This opens up new, systems-level HTS applications for multiplexed libraries, such as identifying transcription factor combinations that induce stem cell differentiation.

In summary, we demonstrated feasibility for developing multiplexed AsCpf1 libraries for functional genomics applications. This approach performed comparably to commonly used monocistronic SpCas9 libraries. A human genome-wide minimized library, Mini-human, was developed based on this strategy and will be available to the research community.

## Methods

**Cell culture**. K-562 cells (ATCC) were cultured in RPMI-1640(Hyclone) supplemented with 10% fetal bovine serum (Gibco, heat-inactivated), penicillin (100U mL$^{-1}$ final concentration) and streptomycin (100U mL$^{-1}$). Lenti-X 293 T (Clonetech) were cultured in DMEM (Hyclone) supplemented with 10% fetal bovine serum (Gibco, heat-inactivated), penicillin (100U mL$^{-1}$ final concentration) and streptomycin (100U mL$^{-1}$).

**Plasmids**. Human codon-optimized AsCpf1 were PCR amplified with primers containing NLS signals in NEB Q5 hot start mastermix from plasmid SQT1659 (Addgene# 78743). lenti-AsCpf1-Blast (Addgene# 84750) was digested with AgeI-HF(NEB) and BamHI-HF(NEB). NLS modified AsCpf1 and digested vector were ligated using Gibson Assembly.

**Immunoblotting**. AsCpf1 antibody (Genetex, GTX133298, 1:250 dilution), Histone H3 antibody (CST, 9715,1:3000 dilution), Beta-tubulin (Sigma, T4026,1:3000 dilution) were incubated overnight at 4 °C. Anti-rabbit (CST,7074,1:2000 dilution) and anti-mouse secondary antibody (CST,7076,1:2000 dilution) were incubated for 3 h at room temperature.

**Immunofluorescence microscopy**. HEK-293T cells were seeded on poly-lysine pre-treated lab-tek chamber slides and transfected with Lipofectamine 3000 per manufacturer's protocol. Phospho-histone H2A.X(Ser139) D7T2V mouse monoclonal antibody (CST, 80312, 1:200 dilution) were used for the primary antibody incubation with manufacturer's standard protocol. Goat anti-mouse Alexa fluor488 (Fisher Scientific, PIA32723,1:600 dilution) were incubated 2 h in 3% BSA PBS-TritonX-100 solution for the detection.

**In-cell western**. HEK-293T cells were seeded on poly-lysine pre-treated 96-well plates with black wall. After treatment, cells are fixed with 4% formalin in PBS. Histone H2A.X(D17A3) monoclonal rabbit antibody (CST,7631, 1:100 dilution) and phospho-histone H2A.X(Ser139) D7T2V mouse monoclonal antibody (CST, 80312, 1:200 dilution) were mixed as a cocktail for in-cell western per manufacturer's standard in-cell western protocol. IRDye 800CW goat anti-mouse IgG (LICOR,925–32210, 1:1000 dilution) and IRDye 680RD goat anti-rabbit IgG (LICOR,925–68071,1:1000 dilution) were used as secondary antibodies followed by Odyssey CLx scanning and signal readout.

**Guide design**. A total number of 342 core-essential genes and 345 non-essential genes were used to generate both Cas9 and AsCpf1-based libraries, with three guides per gene. Design of guides was accomplished by program CLD[36]. On-target selection rules were identical for both Cas9 and AsCpf1 except for the different requirements of PAM. Briefly, guides targeting the most transcripts and are closest to the first exon were prioritized. "NGG" PAM was used for Cas9 and "TTTV" was used for AsCpf1.Bowtie was used for off-target prediction. Mismatch tolerance of Cas9 was set to be up to two mismatches across 20 bp spacer and up to one mismatch across PAM adjacent 18 bp for AsCpf1. Any guides having over two predicted off-target sites across hg19 or targeting regions documented in dbSNP were excluded. One thousand eleven out of 2061 guides were predicted to be off-target free in Cas9 library and 1899 out of 2061 guides were in AsCpf1 library.

**Genome-wide mini-human multiplexed library design**. Guides were identified by adapting the Cas9 library design algorithm developed by Hart et al.[4] Briefly, candidate guide sequences were obtained from exonic regions by scanning the "TTTV" PAM sequence using hg38 and are filtered for homopolymers, and BsmBI restriction sites. Using Bowtie, we aligned the filtered candidate guides across the genome, allowing for one mismatch outside the "TTTV" PAM sequence. Guides

with off-target matches in intronic or exonic regions were excluded, and the remaining guides were ranked based on the number of off-target matches in intergenic regions. A sequence score was assigned to each guide based on the score table presented in Fig. 2a.

**Library construction**. For Cas9-based library, lentiGuide-Puro (Addgene#52963) were digested with FastDigest Esp3I (Thermo Fisher Scientific) and Gibson Assembly was used to clone Cas9-based spacer into the backbone. For AsCpf1-based library, site-directed point mutagenesis was used to incorporate 2 Esp3I (BsmBI) sites flanking Cas9's guide scaffold, generating lentiUniversal-Puro. LentiUniversal-Puro then were digested with FastDigest Esp3I (Thermo Fisher Scientific) and Gibson Assembly was used to clone AsCpf1 mono-cistronic guide into the backbone. For multiplexed AsCpf1 library, guide arrays were cloned into the same backbone with Quickligase ligation kit (NEB). DH10B MegaX (Life technologies) electroporation competent cells were used for transformation. Lenti-X 293 T(Clonetech) were transfected with plasmid library, PMD2.G(Addgene #12259), PsPAX2 (Addgene #12260) to generate lentiviral libraries.

**Library screen**. K-562 cell line was infected with each library at <0.3 multiplicity of infection at aimed coverage of at least 1000-fold. Forty-eight hour post infection, the normal cell culture medium was changed into cell culture medium containing 2 µg mL$^{-1}$ puromycin. 4 days post infection, cells were infected with lentivirus generated from lentiCas9-Blast (Addgene #52962) and lentiAsCpf1–3xMYC correspondingly, about 1000× infected cells were isolated for each library for reference purpose. Six days from the first infection, cells were further selected with 10 µg mL$^{-1}$ blasticidin and split into triplicate for each group. Regular sampling was taken during the process of screening at each time point, cells were pelleted and frozen at −80 °C before further processing. At each sampling and screen process, cells were maintained at a minimum of 1000-fold complexity of the libraries per replicate. For the genome-wide Mini-human library screen in K562, we used a 500× coverage.

**T7E1 assay**. Seventy-two hour post transfection of guide and nuclease, cell genomic DNA was extracted with Qiagen DNeasy blood and tissue kit per manufacturer's protocol. Genomic DNA was used for the template of PCR with NEB Onetaq mastermix with standard buffer per manufacturer's protocol. The product was first denatured at 98 °C for 5 min, then slowly annealed to 75 °C at 1 °C s$^{-1}$ and eventually to 25 °C at 0.1 °C s$^{-1}$. 5U T7 Endonuclease I was used for digestion of less than 300 ng annealed product for 30 min. The digested product was subjected to 2% TAE agarose gel electrophoresis.

**Next-generation sequencing**. Genomic DNA was extracted from cell pellets using DNAzol (MRC Inc.) per manufacturer's protocol. One forth of the total amount of DNA for each sample was used for genomic PCR. Multiplexing barcodes and Illumina sequencing adaptors were incorporated in the 1-step PCR with NEBNext Q5 Hot Start HiFi PCR Master Mix with following conditions: initial denaturing at 98 °C for 1 min, denaturing at 98 °C for 10 s, annealing at 64 °C for 20 s, elongation at 72 °C for 30 s, final elongation for 2 min. PCR cycles for each sample were controlled to the minimal level where the target bands could be seen in 2% agarose TAE gel to ensure unbiased PCR amplification. Target bands were excised from the gel and purified with Freeze 'N Squeeze™ DNA Gel Extraction Spin Columns (Bio-rad), quantified and pooled together. The pooled Illumina library was then subjected to Nextseq550 high output sequencing. Reads were mapped with Bowtie and the mapped reads counts were used for further bioinformatics analysis.

**Low Fat BAGEL**. The BAGEL computational framework estimates the distributions for the core-essential and non-essential genes by bootstrapping the reference genes across 1000 permutations—where roughly 60% of the genes are randomly selected as a training set, and Bayes Factors (BF) for the remaining genes belonging to the testing set are calculated. The final BF for each gene is the average of the Bayes factors obtained in the 1000 permutations. This method, while extremely robust in whole-genome screens, results in overfitting of small library screens such as the benchmark screens we conducted as we are limited to 342 essential and 345 non-essential genes. To address this, we utilized a modified version of BAGEL suited for small libraries, termed Low Fat BAGEL. Rather than focusing on gene level BFs, we leveraged all available data by treating each sgRNA, for each gene, as an independent data point. Similar to BAGEL, 100 permutations are performed on the individual guides of the reference set, rather than on the level of each gene. The 500 guides are bootstrapped across 100 permutations, and the resulting BF for each guide computed is an average of all permutations. An aggregate BF for each gene is then obtained by summing the BFs of individual guides.

**Calculation of AUC for null distributions in the screen**. When a screen with our benchmark libraries cannot distinguish essential gene-targeting guides versus non-essential gene guides completely, the frequency distribution curves of these two categories overlap completely. At any given recall rate, the precision rate should be a constant (i.e., the likelihood of an unknown guide X being an essential gene-targeting guide, which is 0.498). Therefore, for a null distribution, the AUC is 0.498. Lopes et al. for the detailed mathematical proof[37].

## Data availability

All sequencing data that support the findings of this study have been deposited in BioProject with the accession codes "PRJNA483502". The LentiUniversal-Puro and LentiAsCpf1-3xMYC-blast plasmids are available from Addgene with the ID 127749 and 127750 respectively. All other data supporting the findings of this study are available from the corresponding author on request.

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

## Acknowledgements

We thank Draetta lab members for helpful discussions. The human codon-optimized AsCpf1 was cloned from the kind gift plasmid SQT1659 from Keith Joung. Lenti-AsCpf1-Blast was a gift from Hyongbum Kim. LentiUniversal-Puro cannot be constructed without the generous gift LentiGuide-Puro from Feng Zhang. We would like to express our thanks to Florian Heigwer for help with guide design. G.D. and T.H. are supported by MD Anderson Cancer Center Support Grant P30 CA016672. G.D. is supported by NIH/NCI P01 CA117969, NIH/NCI R01 CA218139, the Cancer Prevention and Research Institute of Texas (CPRIT) RP170722, AACR Pancreatic Cancer Action Network 17-65-DRAE, and the Department of Genomic Medicine Sewell Family Chairmanship. T.H. is supported by CPRIT grant RR160032 and NIGMS grant R35GM130119. S.S. was supported by the CPRIT Research Training Grant (RP170067). J.R. was supported by the Altman-Goldstein Discovery Fellowship.

## Author contributions

J.L. and G.D. conceived of and designed the experiments. J.L., C.L., I.H., J.R., M.S., and W.Y. performed the experiments. J.L., S.S., T.H., G.W., and C.B. analyzed the data. The manuscript was written by J.L., S.S., and A.D. with input from all authors.

## Additional information

**Competing Interests:** The authors declare no competing interests.

