## [Peer Review File · Nature Communications]

Reviewers' Comments:

Reviewer #1:

Remarks to the Author:

In this manuscript, Liu et al., used Cpf1 CRISPR nuclease to perform high throughput CRISPR screening. The experimental set up in the manuscript is straightforward. The manuscript is overall well written. The findings in the manuscript are partially expected. The major technical innovation of the manuscript is the demonstration that multiplexed library can perform as well as Cas9 based CRISPR screening, at least at the gene level analysis. Despite the technical novelty, the manuscript is very limited in scope. The current form of the manuscript is only demonstrating the proof of principle that multiplex Cpf1 screening can work; there are no reported biological findings.

Below are some major concerns.

- 1) The current analysis only cursory and is at the library scale. There are no analysis demonstrating the differences and similarities at the individual gene level.
- 2) Although the overall conclusion of the manuscript is that Cpf1 multiplex library works as good as Cas9. The data in the paper is not that convincing. For example, Supplementary Figure 1C shows that Cas9-essential genes are depleted significantly whereas only a limited number of AsCpf1 targeting ones are depleted, Does this show that AsCpf1 still does not work as good as Cas9?
- 3) The authors propose that a whole genome mini library has been constructed. This manuscript needs the actual screening data with this library. Ideally, a whole genome screening needs to be performed with his library and a Cas9 based typical whole genome library. The results need to be compared and contrasted to highlight the advantages and disadvantages of screenings with Cpf1 vs Cas9.
- 4) The multiplex cpf1 screening needs multiple sgRNAs to be expressed in the same cell. To this end, it is possible that multiple double strand breaks are introduced. Multiple DSB may lead to excessive DNA damage and subsequent apoptosis. How does the Cpf1 screening works compare to Cas9 at a multicopy genomic region? Overall the rate of apoptosis and DNA damage level can be quantified.
- 5) I recommend that the method be applied in a biological setting and demonstrate the utility of this approach relative to Cas9. In a given screening, what is the rate of overlap between the top enriched or depleted genes? For example how many of the top 100 or five hundred genes overlap when the screening is performed with cpf1 and Cas9?

Reviewer #2:

Remarks to the Author:

Liu et al developed a new screening platform using arrayed Cpf1 sgRNAs to shrink the size of gene-screening libraries. They provide evidence that a library with 3 sgRNAs arrayed together gives better performance than a single sgRNA Cpf1 library and similar performance to single sgRNA SpCas9 screens, but with fewer overall constructs. Furthermore, they use these screens to determine sequence preferences for Cpf1 sgRNAs, which they subsequently use for designing a "mini-human" library.

Overall, I think this paper has promising results and the screens are well done, but there are some major points that need to be addressed before publication.

Major points:

1. How will off-targets be addressed in a genome-wide screening platform of this type? A single sgRNA construct per gene could provide high false positive rates. Especially in growth screens, can the additional DNA damage be confounding especially if the efficiency of each sgRNA is variable? I believe these are addressable, but I think these caveats should be discussed more thoroughly.

2. I think clearer evidence for synergy between the arrayed sgRNAs is needed. A CRISPRa example is cited, but this may be due to the need to recruit the activation machinery to the genomic locus for longer periods of time. Similar examples have been observed with CRISPRa induced by SpCas9 (PMIDs: 25849900, 23907171), but I think activation is not a good comparison for this.

This evidence is critical as it is not clear the library works better due to the multi-sgRNAs operating synergistically, or if by using three sgRNAs, one is more likely to sample one of the active constructs which might only be 1/3 of the single sgRNAs in the Cpf1 library. If it were the latter, one could argue, it might be possible to pick the best sgRNA for each gene and get the same result without arraying. Perhaps sub-sampling the monocistronic Cpf1 screen to see how using the best sgRNA for each gene would work in comparison to the multi-cistronic would get at this point.

3. Related to point 2: in lines 74-84, more detail should be provided here or in methods regarding exactly which sgRNAs are being used for the single sgRNA screen with Cas9. Are these the best 3 sgRNAs from a previously published library? The worst 3? Any previous data on these? In general most libraries are using 6-10 sgRNAs and perform quite well—the precision-recall curves for this spCas9 library look more comparable to a low-performing early generation library.

Minor points:

Line 106: Define FPR here since you use the acronym later.

Line 115: Have expression levels of Cas9 and Cpf1 been compared here? This may explain the difference in cutting dynamics observed in Fig 1c. Additionally, did the infection of Cpf1/Cas9 work at similar levels? Lower infection rate followed by blasticidin selection may explain some of the differences between Cpf1 and Cas9 temporal dynamics.

Reviewer #1 (Remarks to the Author):

In this manuscript, Liu et al., used Cpf1 CRISPR nuclease to perform high throughout CRISPR screening. The experimental set up in the manuscript is straightforward. The manuscript is over all well written. The findings in the manuscript are partially expected. The major technical innovation of the manuscript is the demonstration that multiplexed library can perform as well as Cas9 based CRISPR screening, at least at the gene level analysis. Despite the technical novelty, the manuscript is very limited in scope. The current form of the manuscript is only demonstrating the proof of principle that multiplex Cpf1 screening can work; there are no reported biological findings.

We appreciate that the reviewer acknowledges the novelty of our research. We agree with the reviewer that this manuscript is technical, and our primary goal for this manuscript is to provide proof of principle that pooled library screenings with a multiplexed AsCpf1 (“Cpf1”) library can be used to execute functional genomics screenings with efficiency comparable to conventional SpCas9 (“Cas9”)-based libraries. Indeed, our manuscript demonstrates that, at the individual guide level, the Cpf1 multiplexed library significantly outperforms conventional Cas9-based libraries and performs equally well at the gene level analysis. We also provide evidence that Cpf1 cuts more slowly versus Cas9. In addition, our manuscript reports new preference rules to design guides for Cpf1 libraries.

To further address her/his concerns, we performed a deeper analysis of our benchmark screening data. We also conducted an additional genome-wide “Mini-human” Cpf1 library screen, as well as cytotoxicity studies to demonstrate that the multiplexed Cpf1 library provided screening benefit compared with Cas9 libraries without sacrificing quality. Details surrounding these efforts are provided below.

Below are some major concerns.

1) The current analysis only cursory and is at the library scale. There are no analysis demonstrating the differences and similarities at the individual gene level.

We agree that our original analysis based on the three benchmark libraries (Cas9-based mono-cistronic, AsCpf1-based mono-cistronic, and AsCpf1-based multiplexed) was at the library level. To compare whether the libraries were comparable at the individual gene level, we performed a gene-matched Wilcoxon test based on the rank of essential genes in three libraries, and we failed to find statistically significant differences ($p=0.2409$ for SpCas9 vs AsCpf1 mono-cistronic, $p=0.2142$ for SpCas9 vs AsCpf1 multiplexed), indicating that Cas9- and Cpf1-based libraries performed similarly at the individual gene level. We also performed a Spearman’s correlation analysis on the z-score of essential genes from benchmark library screen dataset, and we found significant correlations between the SpCas9-based library and the AsCpf1 based libraries (Spearman’s $Rho=0.48$, $p=2 \times 10^{-40}$ for SpCas9 vs AsCpf1 mono-cistronic, Spearman’s $Rho=0.57$, $p=2 \times 10^{-61}$ for SpCas9 vs AsCpf1 multiplexed), (**Rebuttal Figure 1, Supplementary Figure 5a,5b**) again demonstrating high performance similarity between Cas9- and Cpf1-based libraries.

Rebuttal Figure 1 Gene level Spearman's correlation analysis between AsCpf1 and Cas9 based libraries

2) Although the overall conclusion of the manuscript is that Cpf1 multiplex library works as good as Cas9. The data in the paper is not that convincing. For example, Supplementary Figure 1C shows that Cas9-essential genes are depleted significantly whereas only a limited number of AsCpf1 targeting ones are depleted, Does this show that AsCpf1 still does not work as good as Cas9?

In the original manuscript, we showed that, with regard to on-target efficiency, mono-cistronic Cpf1 guide constructs were less active compared to Cas9. However, our data also convincingly demonstrated that, when we concatenated multiple guides together, the efficiency of Cpf1 guides was comparable to Cas9. Moreover, through development and validation of the Cpf1-based libraries, we identified guide design rules to optimize performance, and application of these rules vastly improved efficiency. As a specific example, in our first AsCpf1 benchmark mono-cistronic library, when we randomly picked 10 guides to troubleshoot our first, unsuccessful Cpf1 screening, we found that only 4/10 constructs worked (**Rebuttal Figure 2, Supplementary Figure 3b**). However, when we tested the 16 mono-cistronic AsCpf1 constructs targeting non-essential genes which were derived from the Mini-human library designed using the optimized guide rules – we found that 14/16 constructs worked (**Rebuttal Figure 3, Supplementary Figure 10**).

The performance of a library screen is evaluated based on the width of the assay window; that is, the relative separation between essential genes (positive controls) and non-essential genes (negative controls). The Cas9 library did produce a larger number of depleted essential gene guides, but it did not yield a wider assay window compared to the Cpf1 multiplexed library, which had a higher signal-to-noise ratio. From the precision-recall curves, which are better suited to compare the assay windows versus absolute fold-change levels, it is clear that the Cpf1 multiplexed library outperformed Cas9 at the construct level (**Rebuttal Figure 4, Figure 1e**). The libraries performed similarly well at the gene level, with a slight advantage detected for the Cpf1 multiplexed library based on its slightly slower drop in precision when recall increases (**Rebuttal Figure 5, Figure 1f**). From these analyses, we conclude that that the Cas9 mono-cistronic library screening did result in more significant dropouts, but that the false positive rate of this library was higher versus the Cpf1 multiplexed library, and this did not reflect improved efficiency for on-target effects in the Cas9 screening.

Rebuttal Figure 2 T7E1 assay on 10 randomly selected loci with AsCpf1-nuc and non-optimized guide selection algorithm

Rebuttal Figure 3 T7E1 assay on 16 randomly selected loci with AsCpf1-3xMYC and optimized guide selection algorithm

Rebuttal Figure 4 Construct level precision-recall curves

Rebuttal Figure 5 Gene level precision-recall curves

3) The authors propose that a whole genome mini library has been constructed. This manuscript needs the actual screening data with this library. Ideally, a whole genome screening needs to be performed with his library and a Cas9 based typical whole genome library. The results need to be compared and contrasted to highlight the advantages and disadvantageous of screenings with Cpf1 vs Cas9.

Based on the Reviewer's recommendations, we conducted a genome-wide screen using a "Mini-human" library in K-562 cells using conditions similar to other published screenings in this model using the GeCKOv2 and Wang libraries (Sabatini dataset). Upon comparison with these

two published screenings in Cas9-based libraries, we found that Mini-human performed similarly well.

Rebuttal Figure 6 Core-essential gene overlaps between libraries

Specifically, by applying an FDR threshold of 0.01, we found a similar number (~1200) of hits for both Mini-human (1318) and Wang library (1181). Both libraries identified many more hits (508 and 371) than the GeCKOv2, and the hit concordance rate between any two libraries was similar. Among core essential genes, the percentage overlap between any two datasets was about 50%, which is consistent with a previous report comparing performance among genome-wide libraries³ (**Rebuttal Figure 6, Figure 3a,3b**). We also found similar biological pathways, such as spliceosome, homologous recombination, DNA replication, tRNA biosynthesis, etc. depleted in all three datasets (**Supplementary Figure 6**). Overall, the performance of Mini-human was better compared to GeCKOv2, but it was out-performed by the Wang library (**Rebuttal Figure 7, Figure 3c**). There may be several explanations for the high performance of the Wang library, including its higher screen coverage (1000x, compared to 500x for both Mini-human and GeCKOv2) in cell culture, its highly optimized guides, and its use of ~10 guides per gene. Moreover, in our NGS preparations for the Mini-human screen, we only submitted 1/4 of the total DNA for NGS analysis; thus, a more accurate comparison of coverage with regard to the analyzed screenings is 125x for Mini-human, 500x for GeCKOv2, and 1000x for the Sabatini dataset. The findings from our screening and comparisons with Cas9 libraries is described in greater detail in the revised manuscript.

Rebuttal Figure 7 Precision-recall curve of different CRISPR libraries on K-562

The most advantageous aspect of Mini-human is obviously its compact size. At ~17k constructs, it is <10% of the total size of the gold standard Wang library (~187k) and <15% of the total size of GeCKOv2 (~123k). Even with this drastic reduction in size, its performance is not compromised compared to much larger Cas9 libraries. The compact size facilitates functional genomics research by significantly reducing costs and labor. This is particularly beneficial in cancer drug discovery for multiple reasons, including: (i) the number of models of interest for whole-genome screenings is nearly limitless, (ii) limitations on number of cells for screening in some orthotopic contexts, (iii) need to enable screens in synthetic lethal contexts with standard of care, targeted therapies, and immune-oncology drugs, and (iv) interest in deriving conclusions from screens conducted across multiple genetic contexts to address tumor heterogeneity.

Among the limitations of Mini-human is its PAM, TTTV, which significantly reduces candidate guide sequences compared with Cas9. This could potentially be addressed by using the newly released enAsCpf1-HF1⁴, which has a much broader PAM. A second limitation is that Mini-human, currently, is a two-plasmid system, thus requiring two rounds of virus infection. This was intentional, because it enabled us to test different Cpf1 variants for the screen, and we learned and report in this manuscript that the activity of Cpf1 is crucial for functional genomics screens (we demonstrated that when the efficiency of nuclear localization of Cpf1 efficiency is low, the screening is compromised). We will continue to optimize AsCpf1 for higher activity in our future work.

Cpf1 has been overlooked in the functional genomics arena because Cas9 was released earlier and works well in the in vitro setting. As we describe in the Discussion section of the manuscript, we see great potential for Cpf1-based multiplexed library strategies, especially in the context of combinatorial perturbations and in vivo settings, as well as for CRISPRi/a applications.

4) The multiplex cpf1 screening needs multiple sgRNAs to be expressed in the same cell. To this end, it is possible that multiple double strand breaks are introduced. Multiple DSB may lead to excessive DNA damage and subsequent apoptosis. How does the Cpf1 screening works compare to Cas9 at a multicopy genomic region? Overall the rate of apoptosis and DNA damage level can be quantified.

We would like to clarify that the multiple DSBs induced by our multiplexed Cpf1 constructs are different compared with what has been reported in Cas9 systems where multiple DSBs in highly amplified regions induced apoptosis. In the latter case, Cas9 induces multiple DSBs over very large, multicopy genomic regions of chromosomes, effectively shattering the chromosome. By contrast, the DSBs induced by multiplexed Cpf1 guide arrays are highly localized at the desired locus and, in our multiplexed library, over 90% of genes have guides targeting within 10 kb of the genome. Out of the 687 genes, none of them has guides spanning over 1Mb in the chromosome. (**Rebuttal Figure 8, Supplementary Figure 9**)

Rebuttal Figure 6 DSBs distance distribution in multiplexed AsCpf1 library

Still, the reviewer's point is appreciated and, to address it directly, we designed four multiplexed Cpf1 guide arrays targeting four randomly selected core non-essential genes: ADAM18, IL3, PAX4 and VSX2. These genes were selected because they have been documented to cause no observable growth phenotype upon knockdown or knockout in multiple cell lines in in vitro screens, which makes them suitable for detecting gene-independent cytotoxicity induced by double-strand breaks. In brief, we detected no change in cell viability (CCK8) (**Rebuttal Figure 9, Figure 4a**), apoptosis (annexin V) (**Rebuttal Figure 10, Figure 4c**), or DNA damage (H2AX phosphorylation) (**Rebuttal Figure 11, Figure 4b**) between the multiplexed and mono-cistronic vectors. The details are presented in the revised manuscript.

Rebuttal Figure 7 Relative cell viability with different constructs

Rebuttal Figure 8 Annexin V signal with different constructs

Rebuttal Figure 9 Phosphorylation level of H2A.X with different constructs

5) I recommend that the method be applied in a biological setting and demonstrate the utility of this approach relative to Cas9. In a given screening, what is the rate of overlap between the top enriched or depleted genes? For example how many of the top 100 or five hundred genes overlap when the screening is performed with cpf1 and Cas9?

As mentioned above, we performed a drop-out pooled library screening with our Mini-human library in K562 cells and compared our results with the published K562 genome-wide GeCKOv2 screening dataset and the Sabatini dataset using the Wang library. We observed ~40% overlap for all hits and ~50% concordance for all core-essential gene hits. For all dataset-specific hits, we failed to identify any significantly enriched pathway or biological gene sets using gene ontology analysis for any dataset, indicating similar screening performance of Cpf1 and Cas9. Additional data and discussion of this screening is presented in the revised manuscript.

Reviewer #2 (Remarks to the Author):

Liu et al developed a new screening platform using arrayed Cpf1 sgRNAs to shrink the size of gene-screening libraries. They provide evidence that a library with 3 sgRNAs arrayed together gives better performance than a single sgRNA Cpf1 library and similar performance to single sgRNA SpCas9 screens, but with fewer overall constructs. Furthermore, they use these screens to determine sequence preferences for Cpf1 sgRNAs, which they subsequently use for designing a “mini-human” library.

Overall, I think this paper has promising results and the screens are well done, but there are some major points that need to be addressed before publication.

We appreciate the reviewer's overall favorable assessment of our manuscript. We address points of concern below.

Major points:

1. How will off-targets be addressed in a genome-wide screening platform of this type? A single sgRNA construct per gene could provide high false positive rates. Especially in growth screens, can the additional DNA damage be confounding especially if the efficiency of each sgRNA is variable? I believe these are addressable, but I think these caveats should be discussed more thoroughly.

This is similar to a concern raised by Reviewer #1 (question 4). It is true that, when we multiplex guides, we also increase the likelihood of off-target cutting. In our Cpf1 benchmark and genome-wide libraries, this was mitigated by creating guides using our design rules, as well as the inherently higher precision of Cpf1. We only used the first 18-nucleotide "seed region" sequence to predict Cpf1 off-target sites, and the majority of the guides in our benchmark library (1899/2061) are predicted not to have off-target interaction sites. Also, our multiplexed library had the best performance of the three benchmark libraries we tested. As a final point, the off-target effect of Cpf1 is less pronounced than Cas9 (around one order of magnitude)^{1,2}. Taken together, multiplexing with Cpf1 using fewer than 10 guides should, theoretically, have off-target effects comparable or even favorable to Cas9, and this supported by our data.

As for the potential on-target toxicity introduced by multiplexing different guides targeting the same gene, we would like to clarify that the multiple DSBs induced by our multiplexed Cpf1 constructs are different compared with what has been reported in Cas9 systems where multiple DSBs in highly amplified regions induced apoptosis. In the latter case, Cas9 induces multiple DSBs over very large, multicopy genomic regions of chromosomes, effectively shattering the chromosome. By contrast, the DSBs induced by multiplexed Cpf1 guide arrays are highly localized at the desired locus and, in our multiplexed library, over 90% of genes have guides targeting within 10 kb of the genome. Out of the 687 genes, none of them has guides spanning over 1Mb in the chromosome. (**Rebuttal Figure 8, Supplementary Figure 9**). We designed four multiplexed Cpf1 guide arrays targeting four randomly selected core non-essential genes: ADAM18, IL3, PAX4 and VSX2. These genes were selected because they have been documented to cause no observable growth phenotype upon knockdown or knockout in multiple cell lines in in vitro screens, which makes them suitable for detecting gene-independent cytotoxicity induced by double-strand breaks.

In brief, we detected no change in cell viability (CCK8) (**Rebuttal Figure 9, Figure 4a**), apoptosis (annexin V) (**Rebuttal Figure 10, Figure 4c**), or DNA damage (H2AX phosphorylation) (**Rebuttal Figure 11, Figure 4b**) between the multiplexed and mono-cistronic vectors. The details are presented in the revised manuscript.

2. I think clearer evidence for synergy between the arrayed sgRNAs is needed. A CRISPRa example is cited, but this may be due to the need to recruit the activation machinery to the genomic locus for longer periods of time. Similar examples have been observed with CRISPRa induced by SpCas9 (PMIDs: 25849900, 23907171), but I think activation is not a good comparison for this.

This evidence is critical as it is not clear the library works better due to the multi-sgRNAs operating synergistically, or if by using three sgRNAs, one is more likely to sample one of the active constructs which might only be 1/3 of the single sgRNAs in the Cpf1 library. If it were the latter, one could argue, it might be possible to pick the best sgRNA for each gene and get the same result without arraying. Perhaps sub-sampling the monocistronic Cpf1 screen to see how

using the best sgRNA for each gene would work in comparison to the multi-cistronic would get at this point.

The reviewer is absolutely right, and we have removed language suggesting a “synergistic effect”. We also agree that multiplexing guides into a single vector will increase the chance of sampling an active construct. It is true that, if the optimum guide for each gene were chosen to create a mono-cistronic library, it may be possible for this library to perform similarly to a multiplexed library, especially in our benchmark library setting where all genes result in a very clear growth phenotype when knocked out. However, the benchmark library’s biological utility is quite limited for functional genomics applications, where the goal is to identify genes that yield a previously unknown phenotype within the screening context. In this situation, it is difficult to identify the optimum guide in a high throughput manner a priori, although one could argue that targeted NGS could be used to identify the activity of multiple guides activities in a single run. Multiplexing is an effective approach to reduce library complexity without sacrificing library robustness, and it addresses the issue of inter-cell line heterogeneity such as SNPs which may result in different optimum guide sequences in different screening models.

Our primary goal for this manuscript is to provide proof of principle that pooled library screenings with a multiplexed AsCpf1 (“Cpf1”) library can be used to execute functional genomics screenings with efficiency comparable to conventional SpCas9 (“Cas9”)-based libraries. However, we think it is equally important to inform the scientific community regarding the overall potential of applying this approach, and we discuss in the manuscript that our demonstration is just a first example of where this technology could go.

3. Related to point 2: in lines 74-84, more detail should be provided here or in methods regarding exactly which sgRNAs are being used for the single sgRNA screen with Cas9. Are these the best 3 sgRNAs from a previously published library? The worst 3? Any previous data on these? In general, most libraries are using 6-10 sgRNAs and perform quite well—the precision-recall curves for this spCas9 library look more comparable to a low-performing early generation library.

We wish to clarify that, to make a fair comparison between Cpf1 and Cas9, we did not use a nuclease specific guide optimizing scoring matrix for either enzyme because, when we began our project, there were no optimizing rules for Cpf1. This is why the Cas9 library we built for our study performed below state-of-the-art libraries that have been reported. However, we did apply the off-target filters and on-target rules, such as prioritizing guides closer to 5’ exons. More details regarding guide sequence design can be found in the manuscript in Methods.

Minor points:

Line 106: Define FPR here since you use the acronym later.

Thank you for pointing it out, we have made this correction.

Line 115: Have expression levels of Cas9 and Cpf1 been compared here? This may explain the difference in cutting dynamics observed in Fig 1c. Additionally, did the infection of Cpf1/Cas9 work at similar levels? Lower infection rate followed by blasticidin selection may explain some of the differences between Cpf1 and Cas9 temporal dynamics.

We are unable to directly compare the expression levels of Cpf1 and Cas9 because they share no identical tag or epitopes. Whenever possible, we have controlled to make the systems identical: our Cpf1 and Cas9 vectors share an identical backbone (pLenti-Cas9-blast derived), the viruses are packaged in the same way with the same batches of HEK-293T and culture medium, and nuclease viruses were titered and the multiplicity of infection was the same. We have similarly wondered whether the level of nucleus-localized nuclease may contribute to the temporal dynamics. According to a previous report⁵, the nucleus/cytoplasm ratio of Cas9 varies

from ~0.5 to 3 based on the NLS localization, type and number, while even with a AsCpf1-3xMYC construct, the nucleus/cytoplasm ratio is well below 0.5. However, we do want to clarify that slower cutting kinetics in vitro have been demonstrated with Cpf1 versus Cas9. (**Rebuttal Figure 12, Supplementary Figure 3d**).⁶

Rebuttal Figure 10 AsCpf1 variant's expression level in different cellular fractions

References

1. Kim, D. *et al.* Genome-wide analysis reveals specificities of Cpf1 endonucleases in human cells. *Nature Biotechnology* **34**, 863–868 (2016).
2. Kleinstiver, B. P. *et al.* Genome-wide specificities of CRISPR-Cas Cpf1 nucleases in human cells. *Nature Biotechnology* **34**, 869–874 (2016).
3. Ong, S. H., Li, Y., Koike-Yusa, H. & Yusa, K. Optimised metrics for CRISPR-KO screens with second-generation gRNA libraries. *Sci Rep* **7**, 7384 (2017).
4. Kleinstiver, B. P. *et al.* Engineered CRISPR–Cas12a variants with increased activities and improved targeting ranges for gene, epigenetic and base editing. *Nature Biotechnology* **37**, 276–282 (2019).
5. Suzuki, K. *et al.* *In vivo* genome editing via CRISPR/Cas9 mediated homology-independent targeted integration. *Nature* **540**, 144–149 (2016).
6. Singh, D. *et al.* Real-time observation of DNA target interrogation and product release by the RNA-guided endonuclease CRISPR Cpf1 (Cas12a). *PNAS* **115**, 5444–5449 (2018).

Reviewers' Comments:

Reviewer #1:

Remarks to the Author:

The authors have responded well to my major concerns. I think that the new experimental data together with the text revision have substantially improved the manuscript. Thus, I don't have any further concerns.

Reviewer #2:

Remarks to the Author:

The authors have addressed most of my major concerns, and the addition of a genome-wide screen is helpful. The paper provides some useful information overall and will be a good contribution to the field.

Regarding my previous major point #1, it is nice to see that the authors have added measurements of DNA damage, viability in the presence of introduced Cpf1 sgRNA arrays (though the figure panels in the rebuttal are mislabeled). However, the point I was trying to get at is whether there is additional damage due to the use of multiple sgRNAs. I don't think this should hold up the paper, but it would be very nice to at least see some discussion of the potential for higher levels of off-targets due to the multiple sgRNAs used in these arrays. Because there is no comparison to spCas9 here, it's hard to say whether there is more or less damage than with that system. The authors refer to previous literature indicating an order of magnitude less off-target w/Cpf1, but of course all of these things depend on delivery method, locus targeted, and cell lines used, etc.

Minor (but very important) point: clearer labeling of the figures in 4a-c would be helpful, esp if 'multi' = sg1+sg2+sg3+sg4? Need this info in legends at least, but even better if panels themselves are a bit clearer. For that matter, all figures and legends could use a thorough run-through to improve clarity of labels. Figure 3 is another case where there is really minimal information in the panels and legends. A reader should be able to look at the figure panel and understand what happened, certainly after reading the legend. Describe briefly the experiment done, what #'s are reported, and add labels in the panels so that we can see which # is which.

REVIEWERS' COMMENTS:

Reviewer #1 (Remarks to the Author):

The authors have responded well to my major concerns. I think that the new experimental data together with the text revision have substantially improved the manuscript. Thus, I don't have any further concerns.

Thank you very much.

Reviewer #2 (Remarks to the Author):

The authors have addressed most of my major concerns, and the addition of a genome-wide screen is helpful. The paper provides some useful information overall and will be a good contribution to the field.

Thank you very much.

Regarding my previous major point #1, it is nice to see that the authors have added measurements of DNA damage, viability in the presence of introduced Cpf1 sgRNA arrays (though the figure panels in the rebuttal are mislabeled). However, the point I was trying to get at is whether there is additional damage due to the use of multiple sgRNAs. I don't think this should hold up the paper, but it would be very nice to at least see some discussion of the potential for higher levels of off-targets due to the multiple sgRNAs used in these arrays. Because there is no comparison to spCas9 here, it's hard to say whether there is more or less damage than with that system. The authors refer to previous literature indicating an order of magnitude less off-target w/Cpf1, but of course all of these things depend on delivery method, locus targeted, and cell lines used, etc.

We are sorry that we misunderstood the question. If you were asking, "Will multiplexing of multiple sgRNAs generate de novo off-targets that are not found in the union set of individual sgRNAs," then our short answer is that we do not know. It is indeed a very interesting question, and we truly wish to see this issue addressed. We added brief text in our "Discussion" section of manuscript in response to this query.

Because the in vivo transcription machinery is far from fully understood, the possibility of de novo off-target cuts generated by multiplexing do exist, in our opinion. For example, the short stem loops of multiplexed transcripts might change RNA post-transcriptional machinery, thereby changing the reagent's fidelity. Alternatively, in some sensitive cell lines, multiplexed cutting may have induced drastic changes in the DNA damage response and, therefore, yielded an altered profile of off-target events. For example, induced abortive apoptosis can cause chromothripsis and leads to numerous off-targets detected.

However, we think even if this phenomenon exists, its impact on pooled library screenings will be limited. It will be a more practical and important concern in the context of using multiplexed genome editing therapeutically.

We do agree with reviewer #2's comments "The authors refer to previous literature indicating an order of magnitude less off-target w/Cpf1, but of course all of these things depend on delivery method, locus targeted, and cell lines used, etc.". This was the reason we simply mentioned, "Moreover, the higher

fidelity of Cpf1 compared with SpCas9 contributes to mitigating the increased risk of off-target effects introduced by multiplexing.” Rather than saying there is “no additional off-target effect influencing multiplexed library screen.”

Minor (but very important) point: clearer labeling of the figures in 4a-c would be helpful, esp if ‘multi’ = sg1+sg2+sg3+sg4? Need this info in legends at least, but even better if panels themselves are a bit clearer. For that matter, all figures and legends could use a thorough run-through to improve clarity of labels. Figure 3 is another case where there is really minimal information in the panels and legends. A reader should be able to look at the figure panel and understand what happened, certainly after reading the legend. Describe briefly the experiment done, what #'s are reported, and add labels in the panels so that we can see which # is which.

Thank you very much for pointing this out. We added sentences that are more illustrative and hopefully the clarity is now improved.